# CoSPA: An Improved Masked Language Model with Copy Mechanism for Chinese Spelling Correction

**Shoujian Yang**[1]   **Lian Yu**[1]

[1]School of Software & Microelectronics , Peking University , China

## Abstract

Existing BERT-based models for Chinese spelling correction (CSC) have three issues. 1) Bert tends to rectify a correct low-frequency collocation into a high-frequency one and leads to over-correcting. 2) It fails to completely detect phonic or morphological errors by the current learned similarity knowledge between Chinese characters, and the recall rate still has room to improve. 3) Two-dimensional glyph information of Chinese characters is overlooked and some morphological misused characters may be difficult to detect. This paper proposes a hybrid approach, CoSPA, to address these issues. 1) This paper proposes an alterable copy mechanism to alleviate over-correcting by jointly learning to copy a correct character from input sentence, or generate a character from BERT. No method has used copy mechanism in BERT for CSC. 2) The attention mechanism is further applied on the phonic and shape representation of each character at the output layer. 3) Shape representation is enhanced by mining character glyph with ResNet, and fused with stroke representation via an adaptive gating unit. The experimental results show that CoSPA outperforms the previous state-of-the-art methods on SIGHAN2015 datasets.

## 1 INTRODUCTION

Chinese spelling correction (CSC) is a challenging task in natural language processing (NLP) field, whose target is to detect and correct the misuse of characters or words in Chinese sentences. Existing BERT-based models for CSC have three issues:

**Over-correcting:** Recently, BERT-based non-autoregressive (NAT) language models [Liu et al.,

Table 1: Two examples of Chinese spelling correction.

| Over-correcting example | |
|---|---|
| Input Sentence | ...上网打电动过度不吃饭营养不足导致死亡。 |
| PLOME | ...上网打电话过度不吃饭营养不足导致死亡。 |
| Correct Sentence | ...上网打电动过度不吃饭营养不足导致死亡。 |
| Translation | Excessive playing electronic games on the Internet, skipping meals, and nutritional deficiencies lead to death. |
| Miscorrected example | |
| Input Sentence | ..., 而变成了一个吃香难看的财迷, ... |
| PLOME | ..., 而变成了一个吃香难看的财迷, ... |
| Correct Sentence | ..., 而变成了一个吃相难看的财迷, ... |
| Translation | ..., and become an ugly looking moneygrubber, ... |

2021, Wang et al., 2021, Zhang et al., 2021, Huang et al., 2021] have achieved state-of-the-art performance in CSC task. Such NAT models carry out CSC tasks by predicting based on the whole vocabulary space that can easily lead to miscorrection as the number of vocabularies is large. In addition, they are prone to correct a correct low-frequency collocation into a high-frequency one, resulting in over-correcting. As presented in Table 1, PLOME [Liu et al., 2021] corrects the character '动' (dong, move) to '话' (hua, talk), since the fixed collocation '电话' (phone) has a higher frequency than '电动' (electronic games). CSC is a task where the input and output have the same length with character to character alignment. In some cases, copying is the right action instead of correcting. Although the copy mechanism has been used in sequence-to-sequence framework for CSC, no method has used it in BERT for CSC. This paper proposes an alterable copy mechanism to alleviate over-correcting by jointly learning to copy a correct character from the input sentence, or generate a character from BERT.

**Unable to differentiate a near-phonic and a near-visual conversion:** There are two main types of Chinese character errors: near tone error and near shape error. For errors caused by phonic, such as '香' (xiang, fragrant) and '相' (xiang, observe). However, PLOME is unable to detect correctly with its learned knowledge of the similarity between arbitrary characters, since it may ignore the different importance or relevance in the phonic and glyph aspects of each

*Accepted for the 38th Conference on Uncertainty in Artificial Intelligence* (UAI 2022).

character to CSC task. As such cases have not been detected, the recall rate still has room to improve.

**Overlooking the two-dimensional glyph:** One of the distinctive characteristics of Chinese characters lies in characters both having stroke and two-dimensional graphic information, i.e., the same strokes may constitute different Chinese characters, such as '入' (ru, enter), '八' (ba, eight) and '人' (ren, man), visually similar characters may have different strokes, e.g., '陪' (pei, accompany) and '部' (bu, part). However, recent Chinese pre-training language models with misspelled knowledge overlook the two-dimensional glyph information of Chinese characters, which has visual similarity information for near-visual misused characters.

This paper proposes a model for CSC, called CoSPA, to tackle the above issues, and the contributions are summarized as follows:

- Propose an alterable copy mechanism to alleviate over-correcting by increasing the generation probability of the original character, which is automatically learned by model.
- Introduce an attention mechanism to improve the recall rate, which provides an insight into which aspects of the erroneous input character are more relevant to the correct conversion of the output character.
- Enrich shape embedding by integrating stroke embedding and glyph embedding via an adaptive gating unit.

## 2 RELATED WORK

Early work on CSC followed the pipeline of error detection, candidate generation and selection [Dong et al., 2016]. The neural-based methods have made progress in CSC. Wu et al. took error correction as sequence labeling task with conditional random fields (CRF) [Wu et al., 2018], and Wang et al. treated a CSC task as a sequence labeling problem and used a bidirectional LSTM to predict the correct characters 2018sequence-labeling. This section focus on the models using BERT-based fine-tune, BERT-based pre-trained models with misspelled knowledge, and copy mechanisms.

### 2.1 BERT-BASED FINE-TUNE FOR CSC

FASpell [Hong et al., 2019] first employed BERT as a denoising autoencoder (DAE) for CSC. Recently, more researches fine-tuned BERT-based models using CSC training data. BERT_CRS+GAD [Guo et al., 2021] introduced a BERT with guided replacement strategies of Confusion set, consisting of a number of similar characters sets, to narrow the gap between BERT and misspelling correction, reformed self-attention mechanism to learn the global relationships of the potential correct input characters and the candidates of potentially erroneous characters, and obtained

the rich global contextual information to alleviate the influence caused by error context. Recent researches utilized the external knowledge of character similarity. PHMOSpell [Huang et al., 2021] incorporated both phonological and morphological knowledge from two feature extractors into a pre-trained language model by an effective adaptive gating mechanism. ReaLiSe [Xu et al., 2021] proposed to leverage multimodal information to tackle CSC task, which employed three encoders to learn informative representations from textual, acoustic and visual modalities, and used the selective fusion mechanism to integrate multimodal information. However, these BERT-base models are independently pre-trained from CSC task, thus did not learn any task-specific knowledge during pre-training, which is proved to be most effective [Liu et al., 2021].

### 2.2 BERT-BASED PRE-TRAINED MODELS WITH MISSPELLED KNOWLEDGE

Pre-trained masked language models (MLM) such as BERT [Devlin et al., 2019] and ALBERT [Lan et al., 2019] have set state-of-the-art performance on a broad range of NLP tasks. Different mask strategies enabled models to jointly learn semantics and task-specific knowledge from large scale training data during pre-training.

PLOME [Liu et al., 2021] proposed a pre-trained masked language model with misspelled knowledge for CSC, and put forward the confusion set based masking strategy that randomly replaces 15% of the characters of the input with other characters, where 75% characters from similar character sets in the confusion set (60% phonologically similar and 15% visually similar), and enabled models to jointly learn semantics and misspelled knowledge. It introduced phonic and shape GRU networks to capture phonological and visual similarity features, and is also the first one to introduce pronunciation prediction as an auxiliary objective.

RoBERTa-Pretrain-DCN [Wang et al., 2021] pre-trained the model with the confusion set based masking strategy that randomly replaced 15% of the characters of the input with other characters, where 15% characters from the confusion set. MLM-Phonetic [Zhang et al., 2021] pre-trained a masked language model with phonetic features to improve the model's ability to understand sentences with misspelling and model the similarity between characters and pinyin tokens.

These BERT-based error correction models can be regarded as a limited generation model. They generate a target character from the entire vocabulary space for each character in the input sequence. When the vocabulary space is very large, the probability of miscorrection becomes high. As most of the characters in input are correct in CSC, the high probability leads to over-correcting if no counter-measures are adopted. In addition, some morphological misused characters might

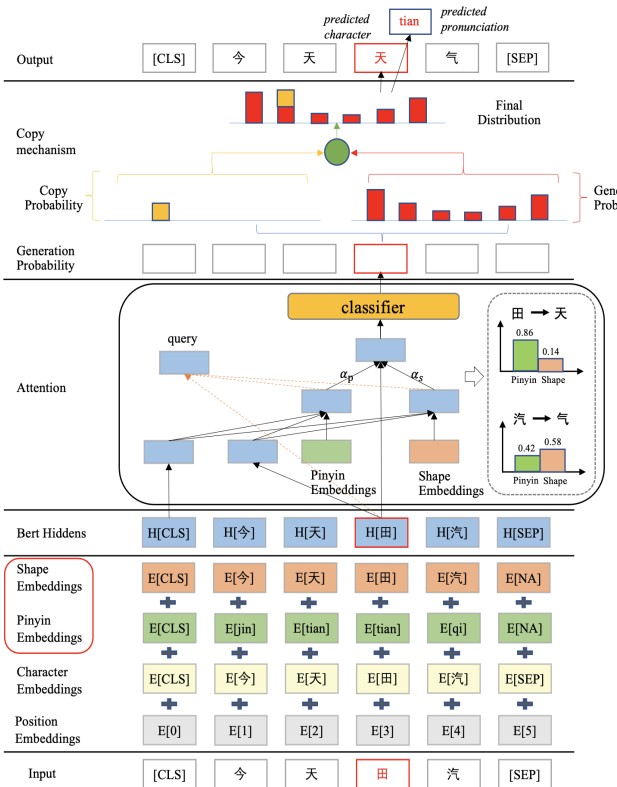

Figure 1: The framework of CoSPA.

be difficult to detect if overlooking the two-dimensional glyph information of Chinese characters.

## 2.3 COPY MECHANISM

Copy mechanism is used in various language generation tasks, such as abstract generation [See et al., 2017], machine translation [Gulcehre et al., 2016], question and answer [McCann et al., 2018], and dialogue [Wu et al., 2019]. It introduces the decoder of seq2seq to improve the performance of the model through "copy and paste" words between inputs and outputs. Wang et al. used a sequence-to-sequence framework with copy mechanism to copy the correction results directly from a prepared confusion set for the erroneous words [Wang et al., 2019]. To the best of authors' knowledge, this paper is the first to leverage a copy mechanism in BERT framework for CSC.

## 3 MODEL

As shown in Figure 1, PLOME [Liu et al., 2021] is used as the base model in this paper. Initially, the input embedding of each character is the sum of character embedding, position embedding, phonic embedding and shape embedding. The hidden states generated by the last layer of transformer encoder attach different importance to phonic or shape em-

bedding by an attention mechanism, then generates the probability from the Softmax output. The final probability of character prediction is the weighted summation of copy probability and generated probability. The examples input contains two types of errors i.e, near-phonic '田' (tian, field) and near-shape '汽' (qi, steam). The phonic and glyph features are introduced to solve the conversion of near-phonic ('田' to '天') and near-shape errors ('汽' to '气'). Given an input sentence $X = \{x_1, x_2, ...x_n\}$, the target is to generate a correct sentence $Y = \{y_1, y_2, ...y_n\}$.

### 3.1 SHAPE EMBEDDING

This paper uses ResNet [He et al., 2016] to encode the character images to get the glyph representations, which has 5 ResNet blocks followed by a layer normalization operation. The glyph representation of $x_i$, $h_i^g$, are defined as follows:

$$h_i^g = LayerNorm(ResNet(I_i)) \tag{1}$$

where $I_i$ is the image of the $i$-th character $x_i$ in the input sentence, and $LayerNorm()$ takes a layer normalization.

The character image of $x_i$ is read from preset font files as shown in Figure 2. Microsoft elegant black in Simplified Chinese is selected, and the size of each character image is set to $32 \times 32$ pixel. To obtain the embeddings of the font in the two-dimensional graphic structures, each block in ResNet halves the width and height of the images, and doubles the number of channels. Thus, the final output is a vector with the length equal to the number of output channels, i.e., both height and width become 1. The number of output channels are set to the hidden size of stroke embedding for the follow-up fusion. The glyph representation of the input sentence is denoted as $H^g = \{h_1^g, h_2^g, ..., h_n^g\}$

Finally, the shape embedding is obtained by an adaptive gating unit served as a gate to finely control the fusion of stroke embedding and glyph embedding. $a_i^{st}$ and $a_i^g$ are the gate values of stroke embedding and glyph embedding, $h_i^{sh}$ is the shape embedding of the $i$-th character $x_i$, and are computed as follows:

$$a_i^{st} = \sigma(W^s[h_i^{st}, h_i^g] + b^s) \tag{2}$$

$$a_i^g = \sigma(W^g[h_i^{st}, h_i^g] + b^g) \tag{3}$$

$$h_i^{sh} = a_i^{st} h_i^{st} + a_i^g h_i^g \tag{4}$$

where $h_i^{st}$ is the stroke embedding of the $i$-th character $x_i$, $W^s$, $W^g$, $b^s$, $b^g$ are learnable parameters, $\sigma()$ is the sigmoid function, and [·] means the concatenation of vectors.

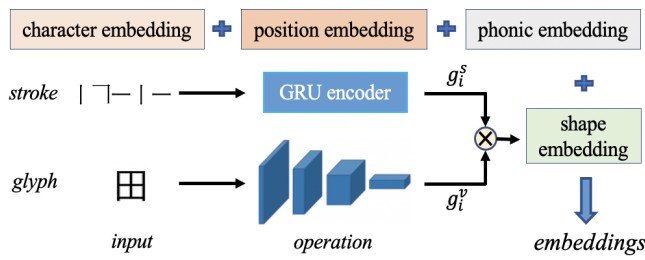

Figure 2: Adaptive fusion of stroke and glyph embeddings

## 3.2 ATTENTION MECHANISM

After the transformer encoder, a 768-dimensional vector representation (donates as the last hidden states of BERT) is output for each position of the input sequence, which is used to perform attention operations on its phonic and shape embedding respectively. Following the use of the [CLS] token to represent the entire sentence [Devlin et al., 2019], in order to consider the semantics of the entire sentence, this paper uses the last hidden state of [CLS] token to perform attention operations at the same time. $F_i$ is the attention vector of the character $x_i$, and defined as follows:

$$F_i = \sum_{k \in \{p,s\}} a_{i,k} E_{i,k} \tag{5}$$

where $a_{i,k} \in R^{1 \times 2}$ is for $i$-th character denoting the corresponding weight of feature $k$ (including phonic and shape), which is computed by

$$a_{i,k} = \frac{1}{2} \sum_{m \in \{h,[CLS]\}} a_{i,k}^m \tag{6}$$

$$a_{i,k}^m = \frac{exp(E_{i,h}^T E_{i,k}/\beta)}{\sum_{k' \in \{p,s\}} exp(E_{i,h}^T E_{i,k'}/\beta)} \tag{7}$$

where $a_{i,k}^m \in R^{1 \times 2}$ denotes the corresponding weights of $m$ (including the last hidden states token and the [CLS] token) to representation $k$ for $i$-th character. $E_{i,h} \in R^{N \times 768}$ (where N is the sequence length) is the hidden states for $i$-th character, $E_{i,[CLS]}$ is the last hidden states for [CLS] token, and $\beta$ is a hyper-parameter determined by experment that controls the smoothness of attention weights. Finally, residual connection is added to $F_i$ and $E_{i,h}$ by linear combination:

$$E_i = F_i + E_{i,h} \tag{8}$$

During the training process, the representation $E_i$ is fed into a fully-connected layer for the final classification. The generated conditional probability $P_{gen}$ of the character predicted for the $i_{th}$ character $x_i$ is defined as:

$$P_{gen}(y_j|X) = softmax(E_i W_c + b_c) \tag{9}$$

where $W_c \in R^{768 \times V}$, $b_c \in R^{768 \times V}$ are learnable parameters for the fully-connected layer, $V$ is the size of the vocabulary and $y_j$ is the predicted j-th character in vocabulary. For more details about the probability of pronunciation prediction, please refer to PLOME [Liu et al., 2021].

## 3.3 COPY MECHANISM

In addition to the original generation probability $P_{gen}$ generated in BERT for each character, a copy probability $P_{copy}$ is added, which is the probability of the character directly output by the model. The final probability is obtained by adding the two probabilities:

$$p = p_{copy} * p_{input} + (1 - p_{copy}) * p_{gen} \tag{10}$$

where $P_{input}$ is one-hot encoding. If the generated top one probability is close to the generation probability of the input character, that usually means model is uncertain to select generated character or copy input character. In this case, the input character is weighted during generation. This procedure is as follows:

$$p_{copy} = \frac{sigmoid(Relu(W_1 h_i)W_2)}{e^{\tau(p_{top1} - p_{in})}} \tag{11}$$

where $W_1$, $W_2$ are trainable parameters of the model, $p_{top1}$ means the generated top one probability of each character, $p_{in}$ means the generation probability of the original input in the vocabulary. $\tau$ is the temperature parameter determined by experment, where increasing $\tau$ makes the distribution flatter. When the difference of the generated top one probability and the generated original input probability is not signifcant, BERT has a low confidence to output top one character, thus tends to output the original chatacter.

## 3.4 LEARNING

The learning process is driven by minimizing negative log-likelihood of the character prediction $L_c$ and phonic prediction $L_d$:

$$L = \alpha * L_c + (1 - \alpha) * L_p \tag{12}$$

$$L_c = -\sum_{i=1}^{n} log p_c(y_i = l_i|X) \tag{13}$$

$$L_p = -\sum_{i=1}^{n} log p_p(g_i = r_i|X) \tag{14}$$

where $L$ denotes the overall objective, $l_i$ and $r_i$ are the true character and phonic for $x_i$, respectively, $g_i$ is the predicted j-th phonic in vocabulary, and $\alpha$ is set to 0.7.

Table 2: Statistics of datasets.

| Training Data | # erroneous sent / sent | Avg.length |
|---|---|---|
| SIGHAN13 | 340/700 | 41.8 |
| SIGHAN14 | 3358/3437 | 49.3 |
| SIGHAN15 | 2273/2339 | 31.3 |
| (Wang et al., 2018) | 271009/271329 | 42.5 |
| Total | 276980/277805 | 42.5 |
| Test Data | # erroneous sent / sent | Avg.length |
| SIGHAN15 | 541/1100 | 30.6 |

## 4 EXPERIMENTS

### 4.1 DATASETS AND IMPLEMENTATION DETAILS

Table 2 shows the statistics of the datasets used in the experiments [Zhang et al., 2020].

**Training Data** The training data is composed of 10K manually annotated samples from SIGHAN [Wu et al., 2013, Yu et al., 2014, Tseng et al., 2015], including 271K training samples automatically generated by OCR-based and ASR-based methods as in [Cheng et al., 2020].

**Evaluation Data** The latest SIGHAN test dataset [Tseng et al., 2015] is used as in [Zhang et al., 2020] to evaluate the proposed model, which contains 1100 testing sentences and half of these sentences included at least one spelling error.

**Evaluation Metrics** Precision, recall and F1 scores are used as the evaluation metrics. Besides character-level evaluation, sentence-level metrics are also adopt for errors detection and correction. These metrics are evaluated using the script from [Cheng et al., 2020].

**Training Details** CoSPA is based on the repository of PLOME [Liu et al., 2021] using Tensorflow1.14 framework. CoSPA is trained using AdamW optimizer for 10 epochs with learning rate 5e-5, a batch size of 32 and a maximum sentence length of 180, and the model is trained with learning rate warming up and linear decay.

### 4.2 COMPARISONS WITH OTHER METHODS

Table 3 shows the evaluation scores compared with all previous methods at detection and correction levels on the SIGHAN2015 test datasets, and CoSPA achieves the new state-of-the-art performance.

Compared with the best baseline method **PLOME**, for character-level and sentence-level, the improvements of CoSPA are 1.3% on detection-level F1 and 1.1% on correction-level F1 respectively.

Both **ReaLiSe** and **PHMOSpell** incorporated phonological

and morphological knowledge into the semantic space for CSC and achieved a relatively good performance. They leverage multimodal information and selectively fuse just to match the characteristics of Chinese characters themselves. CoSPA is more focused on the phonic and glyph aspects of each character to distinguish a near-phonic or a near-visual conversion for the CSC task. CoSPA exceeds both of them illustrates the effectiveness of attention mechanism.

**MLM-phonics**, with the help of additional Pinyin tokens, integrated phonetic features in word embedding, thus increasing the generalization of the model. However, rich shape information of Chinese characters was overlooked. **RoBERTa-Pretrain-DCN** focused on the incoherence problem and modeling the dependencies of the output tokens, since BERT is a non-autoregressive language model, which relies on the output independence assumption. However, they do not model the knowledge of the similarity between arbitrary characters. **BERT_CRS+GAD** narrowed the gap between BERT and spelling error correction with confusion set guided replacement strategy during fine-tuning, but do not learn any task-specific knowledge during pre-training, thus it is sub-optimal.

Specifically, **PN** also employed a Seq2Seq model with copy mechanism, which generated a new sentence considering the extra candidates from confusion set. Instead, CoSPA increases the copy probability of the original input when generating base BERT, and achieves 22.9% F1 improvements at detection-level and 22% F1 improvements at character-level by a large margin. This indicates that the copy mechanism used in BERT has a significant effect.

### 4.3 ABLATION STUDY

As Table 4 shows, when copy mechanism is removed, both the detection and correction F1 scores decrease about 0.8% on character-level and sentence-level. This demonstrates that the copy mechanism makes decoding effective for the CSC task. No matter which component is removed, the performance of CoSPA drops, which fully demonstrates the effectiveness of each part in our model.

**Effect of Character Image Resolution** As Table 5 shows, the performance of the size $64 \times 64$ improved is limited, and the possible reason is that the shape representation usually can be well modeled by strokes.

**Effect of Attention Mechanism** This paper investigates how to better use the attention mechanism for CSC, which is compared against the sum (the summation of hidden states and phonic and shape embedding) and residual connection, with different values of the hyper-parameter $\beta$ in formula (7). The results presented in Table 5 show that the simple sum fails for CSC. It is suggested that both the phonic and glyph features of characters are enhanced without difference,

Table 3: Comparisons among different models in terms of P-R-F (Precision, Recall and F1 score)

| Method | Character-level | | | | | | Sentence-level | | | | | |
|---|---|---|---|---|---|---|---|---|---|---|---|---|
| | Detection-level | | | Correction-level | | | Detection-level | | | Correction-level | | |
| | P | R | F | P | R | F | P | R | F | P | R | F |
| PN | 66.8 | 73.1 | 69.8 | 71.5 | 59.5 | 69.9 | - | - | - | - | - | - |
| ReaLiSe | - | - | - | - | - | - | 77.3 | 81.3 | 79.3 | 75.9 | 79.9 | 77.8 |
| PHMOSpell | - | - | - | - | - | - | **90.1** | 72.7 | 80.5 | **89.6** | 69.2 | 78.1 |
| RoBERTa-Pretrain-DCN | - | - | - | - | - | - | 77.1 | 80.9 | 79.0 | 74.5 | 78.2 | 76.3 |
| MLM-phonetics | - | - | - | - | - | - | 77.5 | **83.1** | 80.2 | 74.9 | **80.2** | 77.5 |
| BERT_CRS+GAD | 88.6 | 87.8 | 88.2 | 96.3 | 84.6 | 90.1 | 75.6 | 80.4 | 77.9 | 73.2 | 77.8 | 75.4 |
| PLOME | 94.5 | 87.4 | 90.8 | 97.2 | 84.3 | 90.3 | 77.4 | 81.5 | 79.4 | 75.3 | 79.3 | 77.2 |
| CoSPA | **95.9** | **88.6** | **92.1** | **98.5** | **85.3** | **91.4** | 79.0 | 82.4 | **80.7** | 76.7 | 80.0 | **78.3** |

Table 4: Ablation results where '- Glyph', '- Attention', and '- Copy' indicate the effects of removing the corresponding mechanisms

| Method | Character-level | | | | | | Sentence-level | | | | | |
|---|---|---|---|---|---|---|---|---|---|---|---|---|
| | Detection-level | | | Correction-level | | | Detection-level | | | Correction-level | | |
| | P | R | F | P | R | F | P | R | F | P | R | F |
| CoSPA | 95.9 | 88.6 | 92.1 | 98.5 | 85.3 | 91.4 | 79.0 | 82.4 | 80.7 | 76.7 | 80.0 | 78.3 |
| - Glyph | 95.7 | 88.4 | 91.9 | 98.2 | 85.1 | 91.2 | 78.9 | 81.9 | 80.4 | 76.4 | 79.7 | 78.0 |
| - Attention | 95.6 | 88.0 | 91.7 | 98.0 | 84.8 | 90.9 | 78.6 | 82.0 | 80.2 | 76.0 | 79.6 | 77.7 |
| - Copy | 95.2 | 87.9 | 91.4 | 97.7 | 84.7 | 90.7 | 78.0 | 81.7 | 79.8 | 75.8 | 79.5 | 77.5 |

which will have a negative impact on the model results, due to the different error types for CSC. The attention mechanism for char, phonics and shape embedding is feasible but is surpassed by the residual connection to hidden states and attention vector for phonics and shape embedding. This indicates that residual connection can provide useful contextual semantic information. Furthermore, hidden state has contained rich representation information. Using it as query can learn the relationship between phonic and glyph features, forcing model to focus on controlling the fusion of the two, which is helpful to identify the conversion type of these typos. A hyper-parameter $\beta$ is incorporated into the attention operation since the dot products may grow large in magnitude, pushing the softmax function into regions where it has extremely small gradients. Based on the experiment results, the Hidden+Attention was chosen with a $\beta$ of 4.

In Table 6 of the weight results, the phonic and glyph representations solve the second issue and have high interpretability. $a_p$ and $a_s$ represent the weights of phonic and shape, respectively. For the phonic and shape error, like '根'(gen, root) and '跟'(gen, and), the first example gives both a higher importance of phonic and shape. For the phonic error, like '师'(shi, teacher) and '书'(shu, book), the second example gives a higher importance of phonic. This shows that the attention mechanism helps model attach different importance to conversion of near-phonic and near-visual. Besides, the average values are calculated of erroneous characters on SIGHAN15, and the phonic and shape weights are 0.73

Table 5: Character-lever F1 with different strategies

| Strategy | D-F | C-F |
|---|---|---|
| Sum | 86.4 | 86.5 |
| Attention($\beta = 1$) | 91.4 | 90.9 |
| Attention($\beta = 4$) | 91.6 | 91.0 |
| Attention($\beta = 10$) | 91.2 | 90.7 |
| Hidden+Attention($\beta = 1$) | 91.9 | 91.2 |
| Hidden+Attention($\beta = 4$) | 92.1 | 91.4 |
| Hidden+Attention($\beta = 10$) | 91.8 | 91.1 |
| Copy($\tau = 0$) | 91.5 | 91.0 |
| Copy($\tau = 1.0$) | 91.9 | 91.2 |
| Copy($\tau = 6.0$) | 92.1 | 91.4 |
| Copy($\tau = 12.0$) | 91.5 | 90.9 |
| image size 32×32 | 92.1 | 91.4 |
| image size 64×64 | 92.1 | 91.5 |

and 0.27, respectively, which means that the phonic aspect is more important than the shape aspect, which is consistent with the fact that the spelling errors caused by near-phonic are more frequent than that by near-shape [Liu et al., 2010].

**Effect of Copy Mechanism** Temperature $\tau$ is a hyper-parameter of neural networks used to control the values of the copy probability by scaling the difference between the generated top one probability of each character and the generated probability of the original input in the vocabu-

Table 6: Attention mechanism weights

| Eng. | Chinese and Japanese. | | | | |
|---|---|---|---|---|---|
| Input | 华 | 语 | 根 | 日 | 文 |
| Output | 华 | 语 | 跟 | 日 | 文 |
| $a_p$ | 0.74 | 0.69 | 0.44 | 0.68 | 0.72 |
| $a_s$ | 0.26 | 0.31 | 0.56 | 0.32 | 0.28 |

| Eng. | And it's hard for teachers to teach. | | | | | | | | | |
|---|---|---|---|---|---|---|---|---|---|---|
| Input | 而 且 老 师 也 很 难 教 师 。 | | | | | | | | | |
| Output | 而 且 老 师 也 很 难 教 书 。 | | | | | | | | | |
| $a_p$ | 0.72 | 0.69 | 0.70 | 0.75 | 0.68 | 0.70 | 0.72 | 0.66 | 0.86 | 0.99 |
| $a_s$ | 0.28 | 0.31 | 0.30 | 0.25 | 0.32 | 0.30 | 0.28 | 0.34 | 0.14 | 0.01 |

Table 7: Comparison of inference time (seconds)

| Method \ Sentences | 5000 | 10000 | 15000 |
|---|---|---|---|
| PLOME | 71.97 | 142.24 | 212.45 |
| CoSPA | 74.60 | 147.73 | 220.80 |

lary. The experiments show that when $\tau = 6.0$, the best performance is achieved.

**Comparison of Inference Time** Table 7 shows the inference times of 5000, 10000, 15000 sentences using PLOME and CoSPA, and CoSPA requires slightly increased times, due to adding more modules, such as copy mechanism.

## 4.4 DEMONSTRATION EXAMPLES

For the false positive case of PLOME, in the first example in Table 8, the ground-truth '悔'(hui, regret) is more suitable according the word '错过' above, which means missing. However, PLOME over-corrects '悔' to '会'(hui, will). Compared with other locations, the difference between the generated top-one probability (0.839) and the generated probability (0.150) of the input character '悔' is not significant. Under these circumstances, the copy probability of the input character should be weighted during generation to output the original character preferentially. After that, CoSPA makes the input character rank as the first.

For the false negative case of PLOME, in the second example, PLOME corrects the erroneous character '素'(su, plain) to '赢'(ying, win). However, according to the word '赔偿' below, which means compensation, CoSPA can correctly consider the different correlation for phonetic error '素' in phonetic, shape and semantic aspects, and predict the correct result '诉'(su, suit).

## 5 FURTHER DISCUSSION

Table 3 gives the aggregated comparison results of CoSPA with PLOME and other approaches in terms of precision,

Table 8: Two examples of inputs and outputs of CoSPA and PLOME

| Over-correcting example | Probability distribution of candidates |
|---|---|
| input | 痛风患者错过后悔哭 | - |
| PLOME | 痛风患者错过后会哭 | 会0.839, 悔0.150, 回0.000, ... |
| CoSPA | 痛风患者错过后悔哭 | 悔0.970, 会0.020, 回0.000, ... |
| Gold | 痛风患者错过后悔哭 | - |
| Trans. | Gout patients miss, regret crying | - |
| Miscorrected example | |
| input | 希望您帮我素取公平，得到他们适当的赔偿。 | - |
| PLOME | 希望您帮我赢取公平，得到他们适当的赔偿。 | - |
| CoSPA | 希望您帮我诉取公平，得到他们适当的赔偿。 | - |
| Gold | 希望您帮我诉取公平，得到他们适当的赔偿。 | - |
| Trans. | I hope you can help me claim justice and get their proper compensation. | - |

recall and F1-score at character level and sentence level, and Table 8 gives two cases that CoSPA is superior to PLOME. Someone might want to know more cases of the performances of these approaches? This section conducts an error analysis on three categories: PLOME is wrong but CoSPA is right, PLOME is right but CoSPA is wrong, and both PLOME and CoSPA are wrong, which accounts for 45.9%, 30.8% and 23.3%, respectively. For each category, two types of incorrect cases are analysed, namely, the false positive and the false negative cases, which affect the precision and recall in CSC, respectively. 10 more examples are randomly selected from each category of each type, which are conducted for 2 runs and the averaged results are as follows.

**Q1: PLOME is wrong but CoSPA is right** As Table 9 shows, for the false positive cases, 20% of them are due to the fact that BERT-base model easily rectifies a correct low-frequency collocation into a high-frequency one, whose standard deviation is 0.14. 25% of them are unable to distinguish 的(de, followed by noun)/地(di, followed by verb)/得(de, followed by adjective) or 他(ta, he)/她(ta, she)/它(ta, it), whose standard deviation (donated as Std Dev) is 0.07, such cases might be handled by Chinese grammar rules. 20% of them are phonetic or shape errors, whose Std Dev is 0.14, a potential approach is to introduce glyph and pinyin features more effectively to break the limitation of artificial confusion sets. 35% of them are other reasons, mainly due to the space of CSC task is very large, the erroneous characters in the real scene are likely to be written incorrectly between any two characters, and the mapping rules between them learned during the training are limited.

As Table 9 shows, for the false negative cases, 25% of them are phonetic or shape errors, whose Std Dev is 0.07. 30% of them are continuous error, whose Std Dev is 0.14, one potential solution is to correct a sentence incrementally through multi-round inference until the model no longer corrects any words, 15% of them are unable to distinguish 的/地/得or 他/她/它, whose Std Dev is 0.07, and 30% of them are other reasons.

Table 9: PLOME is wrong but CoSPA is right

| | |
|---|---|
| **Type 1: False positive cases of PLOME, CoSPA is right.** | |
| **20% of Cases: BERT tends to rectify a correct low-frequency match into a high-frequency.** | |
| Input | 这个宠物死掉的时候有心的养主一定会难过，... |
| PLOME | 这个宠物死掉的时候有心的养生一定会难过，... |
| CoSPA | 这个宠物死掉的时候有心的养主一定会难过，... |
| Gold | 这个宠物死掉的时候有心的养主一定会难过，... |
| Trans | Careful owners will be sad when this pet dies, ... |
| **20% of Cases: Phonetic or shape errors.** | |
| Input | 大家也可怕你的工厂把自然被坏，... |
| PLOME | 大家也可怕你的工厂吧自然破坏，... |
| CoSPA | 大家也可怕你的工厂把自然破坏，... |
| Gold | 大家也可怕你的工厂把自然破坏，... |
| Trans | Everyone is also afraid that your factory destroys nature, ... |
| **35% of Cases: Other reasons.** | |
| Input | ...，不应该都放在各各孩子的身上。 |
| PLOME | ...，不应该都放在个个孩子的身上。 |
| CoSPA | ...，不应该都放在各个孩子的身上。 |
| Gold | ...，不应该都放在各个孩子的身上。 |
| Trans | ..., should not be placed on every child. |
| **Type 2: False negative cases of PLOME, CoSPA is right.** | |
| **30% of Cases: Continuous error.** | |
| Input | 你的工厂机器声音是海曼大声，... |
| PLOME | 你的工厂机器声音是海曼大声，... |
| CoSPA | 你的工厂机器声音是还蛮大声，... |
| Gold | 你的工厂机器声音是还蛮大声，... |
| Trans | Your factory machines are pretty loud, ... |
| **30% of Cases: Other reasons.** | |
| Input | ...，所以我就开发六学有雪方面。 |
| PLOME | ...，所以我就开发留学有学方面。 |
| CoSPA | ...，所以我就开发留学游学方面。 |
| Gold | ...，所以我就开发留学游学方面。 |
| Trans | ..., so I developed the study abroad aspect. |

Table 10: PLOME is right but CoSPA is wrong

| | |
|---|---|
| **Type 1: PLOME is right, false positive cases of CoSPA** | |
| **5% of Cases: Some fixed usages, such as idioms, phrases, and poems.** | |
| Input | ...，不经一番寒澈骨，焉得梅花扑鼻香。 |
| PLOME | ...，不经一番寒澈骨，焉得梅花扑鼻香。 |
| CoSPA | ...，不禁一番寒澈骨，焉得梅花扑鼻香。 |
| Gold | ...，不经一番寒澈骨，焉得梅花扑鼻香。 |
| Trans | without a cold to the bone, how can you get the fragrance of plum blossoms. |
| **50% of Cases: Other reasons.** | |
| Input | 等了半个小时的公车结果看到一辆２９７的公车，... |
| PLOME | 等了半个小时的公车结果看到一辆２９７的公车，... |
| CoSPA | 等了半个小时的公车结果看到一辆１９７的公车，... |
| Gold | 等了半个小时的公车结果看到一辆２９７的公车，... |
| Trans | After waiting for the bus for half an hour, I saw a 297 bus, ... |
| **Type 2: PLOME is right, false negative cases of CoSPA** | |
| **10% of Cases: Lack of world knowledge.** | |
| Input | ...，大家会想到生鱼片、天普罗、寿司之类的东西。 |
| PLOME | ...，大家会想到生鱼片、天妇罗、寿司之类的东西。 |
| CoSPA | ...，大家会想到生鱼片、天普罗、寿司之类的东西。 |
| Gold | ...，大家会想到生鱼片、天妇罗、寿司之类的东西。 |
| Trans | ..., people will think of sashimi, tempura, sushi and so on.. |
| **35% of Cases: Other reasons.** | |
| Input | ...，可是那的时候我有一个重要的考试。 |
| PLOME | ...，可是那个时候我有一个重要的考试。 |
| CoSPA | ...，可是那的时候我有一个重要的考试。 |
| Gold | ...，可是那个时候我有一个重要的考试。 |
| Trans | ..., but I had an important exam at that time. |

Table 11: Cases of both PLOME and CoSPA are wrong.

| | |
|---|---|
| **30% of Cases: Continuous error.** | |
| Input | 我的好朋友，你好！习惯你的生活都很好。 |
| PLOME | 我的好朋友，你好！希惯你的生活都很好。 |
| CoSPA | 我的好朋友，你好！希惯你的生活都很好。 |
| Gold | 我的好朋友，你好！希望你的生活都很好。 |
| Trans | Hello my good friend! Hope your life is good. |
| **25% of Cases: Labeled wrong.** | |
| Input | ...，第一年上办还是没事。... |
| PLOME | ...，第一年上班还是没事。... |
| CoSPA | ...，第一年上班还是没事。... |
| Gold | ...，第一年上半还是没事。... |
| Trans | ..., The first year of work is fine. ... |
| **35% of Cases: Other reasons.** | |
| Input | 我们希望李工厂把这件事如以下处理：... |
| PLOME | 我们希望李工厂把这件事如以下处理：... |
| CoSPA | 我们希望李工厂把这件事如以下处理：... |
| Gold | 我们希望您工厂把这件事如以下处理：... |
| Trans | We hope your factory will deal with this matter as follows: ... |

**Q2: PLOME is right but CoSPA is wrong** As Table 10 shows, for the false positive cases, 5% of them are some fixed usages, such as idioms, phrases, and poems, whose Std Dev is 0.07. A possible approach to handle such cases is utilizing some external knowledge, such as building a collection of special Chinese usages. 20% of them are phonetic or shape errors, whose Std Dev is 0.14. 25% of them are unable to distinguish 的/地/得 or 他/她/它, whose Std Dev is 0.07, and 50% of them are other reasons.

As Table 10 shows, for the false negative cases, 15% of them are phonetic or shape errors, whose Std Dev is 0.07. 10% of them are due to lack of world knowledge, whose Std Dev is 0, it is still very challenging for the existing models to detect and correct such kind of errors, 20% of them are continuous error, whose Std Dev is 0.14. 20% of them are unable to distinguish 的/地/得 or 他/她/它, whose Std Dev is 0.14, and 35% of them are other reasons.

**Q3: Both PLOME and CoSPA are wrong** As Table 11 shows, 30% of them are continuous error, whose Std Dev is 0.14. 25% of them are labeled wrong, whose Std Dev is 0.07, a potential solution is to clean the label of data first, 10% of them are phonetic or shape errors, whose Std Dev is 0. 15% of them are unable to distinguish 的/地/得 or 他/她/它, whose Std Dev is 0.07. and 35% of them are other reasons. There are still many challenge tasks to do in the future.

# 6 CONCLUSION

This paper proposes a hybrid model for CSC, CoSPA, where an alterable copy mechanism is designed to increase the generation probability of the original input to alleviate the over-correcting that BERT-based models widely get into trouble. In addition, to better incorporate phonic and shape of characters, attention mechanism is introduced to provide insight into which features of the input character are more relevant to the correction conversion of the output character. Experimental results on SIGHAN2015 datsets show that CoSPA outperforms almost all the previous state-of-the art methods, demonstrating the effectiveness of the proposed method. Three questions about CoSPA and PLOME that may be concerned are further discussed. The future work will handle the multiple error problem by improving the robustness to noise.

## Acknowledgements

The research was supported by the National Natural Science Foundation of China (No. 61872011).

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
