# OpenReview forum: "CoSPA: An Improved Masked Language Model for Chinese Spelling Correction"
_auai.org/UAI/2022/Conference — UAI 2022 Poster_

### Official Review · Reviewer_YnG2 · 2022-04-10

**Q2(1) Originality/Novelty:** 2
**Q2(2) Significance/Impact:** 2
**Q2(3) Correctness/Technical Quality:** 3
**Q2(6) Clarity Of Writing:** 2
**Q6 Overall Score:** 6
**Q8 Confidence In Your Score:** 4

**Q1 Summary And Contributions:**

This paper presents CoSPA. In order to address the over-correcting issue, CoSPA uses an alterable copy mechanism to decide whether to copy from original sentence or to generate from BERT. For the mis-correcting issue, CoSPA applies attention mechanism on phonic and glyph information. It also uses a ResNet and GRU encoder to encode character image and stroke information respectively. These two representations are then fused with a gating mechanism to help discover near-visual misused characters.


**Q10 Ethical Concerns (Optional):**

No.


**Q2 Assessment Of The Paper:**

More detailed information regarding each of these aspects is given below:

**Q2(4) Quality Of Experiments (Optional):**

3: Good: The experimental evaluation is adequate, and the results convincingly support the main claims.

**Q2(5) Reproducibility:**

3: Good: Key resources (e.g., proofs, code, data) are available and key details (e.g., proofs, experimental setup) are sufficiently well-described for competent researchers to confidently reproduce the main results.

**Q3 Main Strengths:**

1. The Chinese Spelling Correction (CSC) task is of great importance. This paper proposes CoSPA, an improved masked language model for CSC, which incorporates phonic and glyph information to help discover the near-phonic and near-visual misused characters. The idea is solid and related to the conference.
2. The contributions are clearly stated and easy to follow.
3. Experimental results are overall solid and show the effectiveness of each components in the proposed model.


**Q4 Main Weakness:**

1. Other information could also be considered in addition to the glyph and basic pronunciation information mentioned in this paper.
2. Some characters have multiple pronunciations and different corresponding meanings, which brings noise to pinyin representations. It should be better if the authors can analyze the influence of these characters on experimental results.


**Q5 Detailed Comments To The Authors:**

Additional information in Chinese pinyin could also be considered. For example, the pronunciation of Chinese character include five tones (including neutral tone), and the keyboard layout information (e.g. someone may confuse the letter "r" and letter "t" on a standard QWERTY keyboard, which leads to error in pinyin like "run" instead of "tun".)

Is 32*32 enough for some complex Chinese characters to recognize their structures? It is not clear how the resolution of character image affects the performance, and it would be better if the authors could conduct some experiments on it.


**Q7 Justification For Your Score:**

This paper is clearly stated in general, and the contributions are solid and easy to understand and follow. Experimental results show the effectiveness of proposed components.

**Q9 Complying With Reviewing Instructions:**

1: Yes.

---

### Official Review · Reviewer_tQX7 · 2022-04-12

**Q2(1) Originality/Novelty:** 3
**Q2(2) Significance/Impact:** 2
**Q2(3) Correctness/Technical Quality:** 3
**Q2(6) Clarity Of Writing:** 3
**Q6 Overall Score:** 6
**Q8 Confidence In Your Score:** 3

**Q1 Summary And Contributions:**

This paper focuses on chinese spelling correction and overcomes current methods with three shortages by three improvements such as the copy mechanism to alleviate over-correcting, attention mechanism is further applied on the phonic and shape representation of each chacter of the output layer and shape representation is enhanced by mining character glyph with ResNet. Experiments on SIGHAN2015 show tha the proposed approach outperforms sota methods.

**Q2 Assessment Of The Paper:**

More detailed information regarding each of these aspects is given below:

**Q2(4) Quality Of Experiments (Optional):**

3: Good: The experimental evaluation is adequate, and the results convincingly support the main claims.

**Q2(5) Reproducibility:**

3: Good: Key resources (e.g., proofs, code, data) are available and key details (e.g., proofs, experimental setup) are sufficiently well-described for competent researchers to confidently reproduce the main results.

**Q3 Main Strengths:**

1. the three ideas for improving chinese spelling correction are interesting and effective;
2. the question of chinese spelling correction itself is a good direction of improving bert-style models;
3. better results are achieved by the method than sota methods.

**Q4 Main Weakness:**

1. prefer to know details of inference time;
2. not quite sure if the scalability of the method described in this paper. prefer detailed discussions about this direction.

**Q5 Detailed Comments To The Authors:**

1. any detailed comparison of inference speed?
2. in Table 3, not quite understand in "sentence-level, detection-level", CoSPA achieved 79.0 yet baseline PHMOSpell achieved 90.1. Any ideas of improve these scores?
3. is your method applicable to other languages such as Japanese? what should be prepared for other language's spelling correction?

**Q7 Justification For Your Score:**

1. the direction of chinese spelling correction and the three novel parts for improving the accuracies are solid and with rich examples;
2. rich baselines are adapted in this paper for comparison.

**Q9 Complying With Reviewing Instructions:**

1: Yes.

---

### Official Review · Reviewer_fqEc · 2022-04-15

**Q2(1) Originality/Novelty:** 3
**Q2(2) Significance/Impact:** 3
**Q2(3) Correctness/Technical Quality:** 3
**Q2(6) Clarity Of Writing:** 3
**Q6 Overall Score:** 6
**Q8 Confidence In Your Score:** 4

**Q1 Summary And Contributions:**

This paper proposes a hybrid method to improve Chinese spelling correction. To alleviate the over-correction problem, this paper designs an alternative copy mechanism to directly copy Chinese characters from the input. To better model the relevance between the features and the output, this paper further presents an attention mechanism to incorporate both phonic and shape features of Characters. Experiments show that the proposed hybrid method is effective.

**Q2 Assessment Of The Paper:**

More detailed information regarding each of these aspects is given below:

**Q2(4) Quality Of Experiments (Optional):**

3: Good: The experimental evaluation is adequate, and the results convincingly support the main claims.

**Q2(5) Reproducibility:**

3: Good: Key resources (e.g., proofs, code, data) are available and key details (e.g., proofs, experimental setup) are sufficiently well-described for competent researchers to confidently reproduce the main results.

**Q3 Main Strengths:**

S1: The motivation of this paper is quite convincing and the proposed method is simple yet effective.
S2: The analysis is comprehensive and the conclusions are well supported.
S3: This paper is clearly written and easy to follow.

**Q4 Main Weakness:**

W1: The proposed method only fit the scenario where the output is the same length as the input. It is interesting to see whether duplication and omission problems in CSC can be solved.
W2: In the analysis part, different Chinese errors are displayed with tables but the specific error type and corresponding English explanations are not provided in the table, making it less readable for non-Chinese speakers.

**Q5 Detailed Comments To The Authors:**

This paper proposes a hybrid method to improve Chinese spelling correction. To alleviate the over-correction problem, this paper designs an alternative copy mechanism to directly copy Chinese characters from the input. To better model the relevance between the features and the output, this paper further presents an attention mechanism to incorporate both phonic and shape features of Characters. Experiments show that the proposed hybrid method is effective.

Strengths:
S1: The motivation of this paper is quite convincing and the proposed method is simple yet effective.
S2: The analysis is comprehensive and the conclusions are well supported.
S3: This paper is clearly written and easy to follow.

Weaknesses:
W1: The proposed method only fit the scenario where the output is the same length as the input. It is interesting to see whether duplication and omission problems in CSC can be solved.
W2: In the analysis part, different Chinese errors are displayed with tables but the specific error type and corresponding English explanations are not provided in the table, making it less readable for non-Chinese speakers.

**Q7 Justification For Your Score:**

The motivation of this paper is quite convincing and the proposed method is simple yet effective.

**Q9 Complying With Reviewing Instructions:**

1: Yes.

---

### Decision · Program_Chairs · 2022-05-15

**Decision:**

Accept (Poster)

**Comment:**

Meta Review: This paper identifies issues in existing BERT-based method for the Chinese spelling correction problem (CSC) and proposes ways (e.g., an alternative copy mechanism and an enhanced shape representation) to address this. The reviewers consistently found the problem of much importance and interest, and the proposed solution valuable and effective. I would strongly encourage the authors to incorporate the suggestions from the reviewers (e.g., inference time comparison, more explanation of error cases) when preparing a revised version of this paper.